# Beam Formation and Vernier Steering Off of a Rough Surface

**DOI:** 10.3390/mi12080871

**Published:** 2021-07-24

**Authors:** Eric K. Nagamine, Kenneth W. Burgi, Samuel D. Butler

**Affiliations:** Air Force Institute of Technology, 2950 Hobson Way, Dayton, OH 45433, USA; eric.n.writes@gmail.com

**Keywords:** diffuse, scattering, enhancement, phase modulation, spatial light modulator

## Abstract

Wavefront shaping can refocus light after it reflects from an optically rough surface. One proposed use case of this effect is in indirect imaging; if any rough surface could be turned into an illumination source, objects out of the direct line of sight could be illuminated. In this paper, we demonstrate the superior performance of a genetic algorithm compared to other iterative feedback-based wavefront shaping algorithms in achieving reflective inverse diffusion for a focal plane system. Next, the ability to control the pointing direction of the refocused beam with high precision over a narrow angular range is demonstrated, though the challenge of increasing the overall scanning range of the refocused beam remains. The method of beam steering demonstrated in this paper could act as a vernier adjustment to a coarse adjustment offered by another method.

## 1. Introduction

Diffuse scattering of light challenges traditional imaging systems, which rely on ballistic light to operate. When light propagates through turbid media, or reflects off an optically rough surface, it scatters diffusely. Since 2007, new imaging techniques have been able to operate under non-ideal conditions with diffuse scattering by incorporating a process called Wavefront Shaping to their imaging processes. Wavefront Shaping is a family of techniques that compensate for diffuse scattering through the use of spatial light modulators (SLMs) or Digital Micromirror Devices (DMDs) to modify the spatial distribution of amplitude or phase of incident wavefronts.

Mathematical models for wavefront shaping treat rough surfaces as a complex field that interfere with incident light [1]. Controlling the resultant reflected wavefront can be achieved by modulating the incident light. Properly modulated incident light has been shown to refocus after interfering with scattering surface or turbid media in a process dubbed inverse diffusion. Numerous feedback-based algorithms and approaches such as matrix measurements have been developed that determine the required modulation to refocus light after transmissive and reflective scattering [2,3,4,5,6,7,8,9,10,11,12,13,14]. In particular, Vellekoop provides a detailed summary of several different approaches taken [11].

Wavefront Shaping is distinguished from existing free-space adaptive optics due to the use of turbid media or rough surfaces. Depending on the application, the scattering medium may also be static relative to the time required to find an adaptive optics solution. Existing wavefront shaping techniques have been applied to microscopy, tomography, fiber optics, super-resolution imaging, and a growing number of other use cases [11,15,16,17,18,19]. Applications of wavefront shaping specifically involving reflection from a rough surface as opposed to transmission through turbid media are not as numerous in the literature, but there is a continuing interest to utilize reflective inverse diffusion to image around corners [9,12,13,14,20,21]. There also could exist unexplored use cases for wavefront shaping in LiDAR with certain limitations. The design of LiDAR systems is trending toward miniaturization, which may necessitate integrated photonics or other miniaturized technologies that preclude liquid crystal SLMs or MEMS mirror technology [22], which are uniquely capable of displaying and rapidly updating phase masks required for wavefront shaping. Nevertheless, very large optical phased arrays could potentially perform the function of wavefront shaping.

One method that uses wavefront shaping principles to image around corners is to modulate light diffusely reflected from a rough surface to reestablish the light’s ballistic path. This method depends on already present illumination in the scene, and it enables imaging as if the reflected light had specularly reflected from a smooth surface. Previous demonstrations of this technique required calibration sources or guide stars in the around-the-corner scene, although techniques are still being investigated that do not require prior knowledge of the calibration source [9,23].

The scenario when imaging occurs without access or prior knowledge of the scene is sometimes called indirect imaging or non-line-of-sight imaging. A subset of indirect imaging methods require that illumination be actively provided to the scene. These active methods depend on some intermediate surface that has a direct line of sight to the scene and can reflect illumination into it. The intermediate surface functions as either a virtual illumination source, a virtual detector, or both [12,13,14,24,25]. An indirect imaging system with these imposed limitations must be able to rapidly create and then control a virtual illumination source at any surface, even in the presence of noise. This motivates the need for reflective inverse diffusion to refocus a scattered beam to be used as a virtual illumination source in such a system. Additionally, the system must be able to steer the refocused beam onto the target scene.

## 2. Refocusing Reflected Light from an Intermediate Surface

### 2.1. Materials and Methods

The laboratory setup in the experiment was called a focal plane setup, shown in Figure 1. Previous works discuss the merits of different optical setups for achieving reflective inverse diffusion [12,13,20]. The optical source is a frequency-stabilized CW laser at 632.8 nm, expanded and collimated to fill the SLM at near-normal incidence. The SLM imparts a phase mask on incident, collimated light. That light, still approximately collimated, is then focused onto a sample scatterer. The scatterer is placed at the beam waist and is angled to separate incident and reflected light. The Charge Coupled Device (CCD) is placed at the center of reflection’s main specular lobe. The SLM used was a Meadowlark Optics Reflective XY series model P512 SLM, which has 512×512 square pixels, with a pixel pitch of 15μm for a total effective aperture 7.68 mm × 7.68 mm. The device has a reported fill factor of 83.4%. The device controller was a PCIe 16-bit controller providing 16-bit phase discretization between 0 and 2π; however, a manufacturer-provided lookup table provided correct digital to phase outputs for 16,384 levels. The detector used was a Thorlabs model 4070M-GE-TE, which has 2048×2048 square pixels, with a pixel pitch of 7.4μm. Both devices were controlled by a desktop computer in Matlab^®^ using manufacturer-provided libraries. The scattering samples used were polished Nickel and white paint on glass. Surface properties are listed in Table 1. As a metal, the polished Nickel sample was a pure surface scatterer, as no transmission of light into the medium occurred, whereas transmission did occur for the white paint sample.

Iterative feedback-based wavefront shaping algorithms were used to achieve reflective inverse diffusion. In these algorithms, the SLM was partitioned into N×N square segments. As in previous work, it was assumed that the scattering sample would act as a lens with focal length $Z_{2}$ after optimization and that the minimum spot size at the detector would be based on the diffraction limited beam waist of a single SLM segment propagating through the system [12]. In that work, it was shown that the spot size at the detector would be inversely proportional to *N*. The detector pixels were accordingly partitioned into square channels such that the width of each channel was as close as possible to the diffraction limited beam waist for the chosen *N*.

Prior to any wavefront shaping, the reflection of the light off of the sample scatterer created a diffuse speckle pattern centered around the main specular lobe of reflection from the sample surface. A single detector channel was chosen to be the target of the refocused, optimized beam, and the remaining channels measured the background speckle intensity over the extent of the detector. The metric used to measure the efficacy of the wavefront shaping algorithm was termed enhancement, η, and is defined in Equation (Equation 1) as
(1)$$\eta \equiv \frac{I_{opt}}{\langle I_{ref}\rangle },$$
which is the ratio of the optimized beam intensity over the averaged background speckle intensity [2]. Enhancement measures the improvement in selected directivity of energy similar to directional gain in antennas. The maximum enhancement for inverse diffusion with an SLM is proportional to the number of SLM segments used [2,12]. There is an additional effect based on the number of allowed phase levels. Allowing more phase levels allows for greater phase resolution of the SLM phase masks and increases maximum enhancement [12].

Three algorithms were adapted from the existing literature. These algorithms were the partitioning algorithm (PA), the continuous sequential algorithm (CSA), and a genetic algorithm (GA) [3,8]. The process of feedback in these iterative feedback-based wavefront shaping algorithms is to set a test phase mask on the SLM and to make an intensity measurement on the detector to calculate the resultant enhancement. A modification of the SLM phase mask creates a change in the enhancement, which has either increased or decreased the optimization of the phase mask. Each iteration begins with a current best guess for the SLM phase mask that is optimized to maximize enhancement. The algorithms select a sub-set of the phase mask’s segments and apply different phase levels to that subset in an attempt to find a phase mask with higher enhancement than the current best guess. If any of the tested phase masks create higher enhancement than the current best guess, the best guess is updated.

The iterative feedback-based wavefront shaping algorithms begin with initial unoptimized phase masks and iteratively modulate the phase of selected SLM segments while measuring the enhancement to find the ideal phase levels for the selected phase masks. The continuous sequential algorithm (CSA) modulates single SLM segments sequentially through the discrete range of allowed phase levels. In this experiment, only 15 phase levels were allowed, and the number of segments over which the SLM optimized were limited to 32×32. Pre-optimization of the CSA is performed by first running the algorithm on larger super-segments, beginning with a partion of the SLM into 2×2 segments and then 4×4 and continuing to divide the partitions until until the SLM is subdivided into 32×32 segments. In a noiseless system, the CSA would eventually converge on the true solution, but as the proportion of pixels being modulated decreases, relative intensity noise (RIN) of the laser obscures the intensity modulation caused by the modulation of selected SLM segments, degrading the ability of the algorithm to determine the optimal phase level for the segments. In the experiment, the segment size used by the CSA was 16×16 SLM pixels and the detector channel size was accordingly 8×8 pixels.

The partitioning algorithm (PA) operates similarly to the CSA, except that in each iteration, it chooses a random partition of the SLM’s segments instead of following a sequence. It cycles through the 15 allowed phase levels for all segments in the chosen partition simultaneously. Variations in the PA involve a decrease in the partition size as enhancement increases so that smaller and smaller changes are made. As in the CSA, RIN limits the effectiveness of the PA when small partitions are chosen. In the experiment, the segment size used by the PA was 4×4 SLM pixels, and the detector channel size was 2×2 pixels.

The GA does not operate based on of a deliberate sequential search in either the segments or the possible phase values for each segment. Instead, it maintains a population of phase masks and splices them together in random ways each iteration while also adding a small amount of randomized mutations. The GA can operate in arbitrarily large search spaces without being slowed down by a sequential search. The GA was allowed to choose from all 16,383 possible phase levels of the SLM and used a segment size of 4×4 SLM pixels with a corresponding detector channel size of 2×2 pixels. Regarding the RIN challenge, the GA is guaranteed a minimum amount of variability between phase masks, but RIN may still degrade the algorithm’s ability to determine relative optimality between similar phase masks.

### 2.2. Results and Discussion

The genetic algorithm proved to be the most robust. Figure 2 shows the per-iteration comparison of enhancement for the three algorithms when applied to both samples. One qualitative benefit of the GA is that because it does not perform a sequential search over segments or phase, its running speed is agnostic to the size of the search space. This is an important feature because any algorithm that depends on a sequential search over segments or phase levels will require more intensity measurements to cycle through each segment of the SLM phase mask when spatial or phase resolution of the SLM phase masks are increased. It was for this reason that the GA and PA were allowed to have a greater number of smaller SLM segments than the CSA—there is no penalty for doing so.

The performance of all algorithms was limited by relative intensity variations present in the system, primarily the laser RIN. The CSA in particular was affected because of how it modulates only one SLM segment at a time, changing only a small fraction of the overall wavefront resulting in a small intensity modulation. The PA and GA were able to succeed because a much larger percentage of the overall SLM was modulated for each measurement. This allowed those algorithms to more reliably differentiate the optimality of different SLM phase masks.

## 3. Non-Mechanical Beam Steering

### 3.1. Materials and Methods

Non-mechanical beam steering exploited the Fourier transform relationships inherent to the focal plane geometry of the optical setup in order to create a phase tilt at the scatterer. The field at the focal plane of a lens is a coordinate-scaled Fourier transform of the wave incident on the lens [26]. For some function g(x) that has a Fourier transform F[g(x)], the shift theorem of Fourier transforms states that
(2)$$F\left[g\left(x-\Delta x\right)\right]=F\left[g\left(x\right)\right]\exp\left[-i2\pi \left(f_{x}\Delta x\right)\right],$$
where Δx is the size of the shift. Simulations have shown that circularly shifting the SLM phase mask approximates the required shift needed to create phase tilts at the scatterer for small shifts [21]. However, these shifts vacate the optimized phase mask from the SLM and leave behind unoptimized pixels. Therefore, there is a maximum distance the SLM phase mask can be shifted before the entire SLM is unoptimized. A circular shift was chosen because it reuses the portion of the optimized SLM phase mask that is shifted off of the boundary of the SLM and places them in the vacated pixels. This is advantageous because the vacated pixels would otherwise contribute to background speckle but are instead optimized to focus the light toward a secondary spot. Because of the nature of circular shifts, this is also disadvantageous because the secondary spot will be equally optimized with the desired spot when the SLM phase mask is circularly shifted by 50%, after which it is more optimized. Figure 3 shows additional simulated data demonstrating the creation of phase tilts at the scatterer’s surface that are superimposed over the wavefront-shaped field due to circular spatial shifts of the SLM’s phase mask. The slope *m* of the phase tilts at the scatterer can be derived from Equation (Equation 2). Extracting the slope from the phase tilt term in Equation (Equation 2) and applying appropriate coordinate transformations from Fourier optics,
(3)$$\begin{array}{cc} m & =-\frac{2\pi }{\lambda f}\Delta x\;\;\;\;\;\left(\frac{radians}{m}\right);\\ \; & =-\frac{\Delta x}{f}\;\;\;\;\;\left(unitless\right), \end{array}$$
where f is the focal length of $L_{1}$ and λ is the wavelength. The angle of this tilt, which shall be denoted $\theta_{scan}$, is given in Equation (Equation 4) as
(4)$$\theta_{scan}=\tan^{-1}\left(m\right).$$

### 3.2. Results and Discussion

Figure 4 and Figure 5 show the relative reduction in enhancement of the beam at the detector as a function of the scan angles in two dimensions while Figure 6 demonstrates the individual relationships between horizontal and vertical scans and enhancement. These data are averaged over multiple trials where the GA was optimized to different initial detector channels. The SLM phase mask was shifted in 16-pixel increments, which is 240μm on the detector. The resultant scan angle for each shift was calculated trigonometrically from the displacement of the refocused beam on the detector. The beam’s displaced position was determined by centroiding a windowed region of the detector around the beam. Coarse displacement estimates derived from Equation (Equation 4) set the window boundaries. Table 2 shows the modeled and experimentally observed scan angle increments due to incremental shifts of 240μm of the SLM phase mask.

The nickel sample experiences a quasi-linear decrease in enhancement as a function of scan angle over the range of angles tested. The scan angle at which the enhancement decreases by 50% along either the the horizontal or vertical axis is approximately 0.2°±0.02°. The paint sample is notably asymmetrical between the axes. Horizontal scans on the paint sample appear to experience a quasi-linear decrease in enhancement over the range of scan angles tested; however, it is clear from Figure 6 that the behavior of vertical scans cannot adequately be described with a linear relationship over the entire range of angles tested. In the paint sample, scans drop the enhancement to 50% at 0.033°±0.015° along the horizontal axis and by 0.034°±0.017° along the vertical axis. The subsequent decrease by 75% occurs at 0.22°±0.02° along the horizontal axis and at 0.15°±0.02° along the vertical axis.

The likely explanation for the difference between the behavior of the nickel and paint samples is that most of the reflection from nickle sample is singly scattered from the surface while only a small fraction of the reflection from the paint is singly scattered. The phase tilts of the electric field incident on the paint sample are not likely to be maintained for the portion of light transmitted past the paint’s surface due to the multiple scattering that occurs. This would account for the seemingly discontinuous dropoff that occurs when any shift is made from the center, as seen in Figure 5. With this in mind, it is possible that the overall trends shown in Figure 6c,d are only attributable to the singly scattered light from the paint sample. The other notable feature of the paint sample is that vertical scanning has a steeper decline than horizontal scanning. This was not due to any non-uniformity of the paint but must be attributed to the fact that the horizontal plane is the plane of incidence. Additionally, the central cross sections in Figure 6 show that the largest relative decrease occurs after the first circular shift, both horizontally and vertically. Using this view of the data, it is easy to see the discontinuous dropoff in the paint as well as a larger than average dropoff in the metal sample.

The likely explanation is that the beam steering technique only holds for singly scattered light. The relative enhancement for the central cross section below that of the other cross sections is then attributable to some function of the proportion of singly scattered light and multiply scattered light in addition to the relative optimality of the phase mask. As a surface reflector, the metal sample should cause less multiple scattering than the paint sample. This is believed to be why the decreases in relative enhancement for the metal sample’s central cross section are less prominent than the paint sample’s.

The presented technique for beam steering could provide a vernier beam steering capability but would need to be paired with another technique if scanning is required over large solid angles, such as in LiDAR. Additionally, the technique is highly dependent on the focal plane setup, which places design constraints on the use cases. For instance, it would not be practical to use this technique with an OPA in LiDAR because the OPA itself could act as a steerable illumination source. Other LiDAR illumination techniques, such as flash LiDAR, utilize diffractive elements such as metasurfaces to create point clouds [22,27]. At a minimum, there is a possibility of using a focal plane system with these approaches when the diffractive element can be separated from the laser. The goal of such applications could be to perform vernier rotations of the point cloud.

## 4. Conclusions

The GA is able to optimize the SLM phase mask with relative speed in laboratory conditions that offer stability. However, it is unclear what level of optical power or enhancement of the refocused beam that is needed to meet the requirements for indirect imaging. Further analysis including link budget analyses of possible scenes is needed to determine the level of enhancement that is required of the wavefront shaping algorithm. Independent of algorithm improvements, advances in SLM technology to support faster update speeds will proportionally speed up the wavefront shaping process, and the availability of low RIN lasers can also improve algorithm performance.

The SLM circular-shift method of beam steering is able to create small changes in the pointing direction of refocused beams. For this experimental geometry, circular shifts could be made to modify the pointing direction in increments as little as several thousandths of a degree. However, this method suffers large decreases in efficiency when applied to white paint, which is a bulk scatterer. The nickel sample fared better, likely because it is a surface scatterer. Depending on what level of losses are acceptable, the circular shift method could be used as a vernier beam steering adjustment to correct the overall scan angle to a precision below a tenth of a degree for reflective surfaces down to several thousandths of a degree. A method for coarse adjustments of the scan angle to accommodate a wider field of view is still required.

It was demonstrated that a genetic algorithm may be the best choice of all iterative algorithms for maximizing the enhancement of refocused beams in reflective inverse diffusion. It was also demonstrated that circular shifts of the SLM phase mask in a focal plane system can be used for high precision vernier adjustments of a refocused beam’s pointing direction for a surface scattering sample. The circular shift method suffered large losses with the bulk scattering sample and may only be suitable for surfaces with high surface reflection. Finally, a method for wide field of view beam steering of the light in reflective inverse diffusion must still be discovered.

## Figures and Tables

**Figure 1 micromachines-12-00871-f001:**
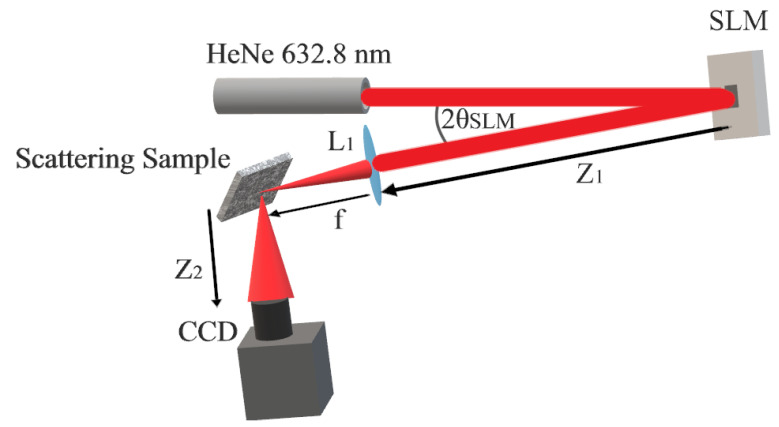
Simplified focal plane setup used in reflective inverse diffusion experiments. Polarized, collimated light is expanded and incident on the SLM at a slight angle. Light reflected from the SLM is focused by lens $L_{1}$ onto the scattering sample and reflected to the CCD. In the lab, the incident angle of the beam onto the SLM, denoted $\theta_{SLM}$, was 3.98°±0.01°, while $Z_{1}=54\pm 0.5$ cm, $Z_{2}=40\pm 0.5$ cm, f=400 mm.

**Figure 2 micromachines-12-00871-f002:**
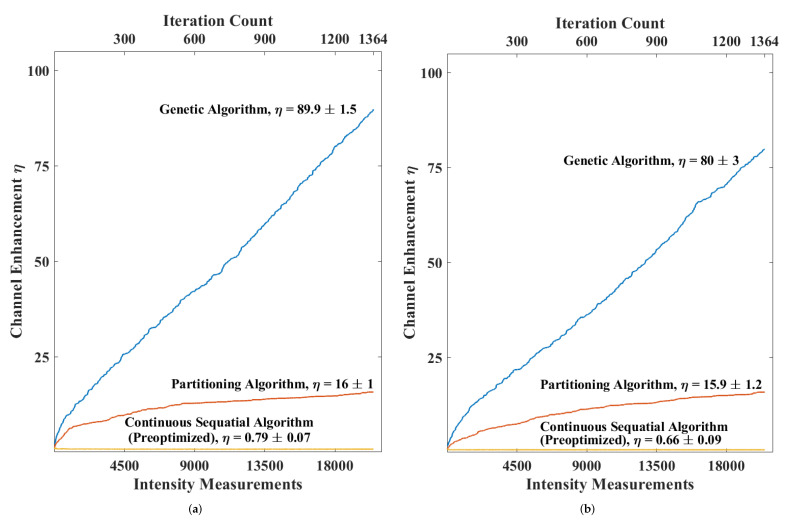
Enhancement achieved using different techniques for reflective inverse diffusion. In (**a**), the performance on 600 grit polished nickel is shown. In (**b**) the performance on zinc oxide white paint is shown.

**Figure 3 micromachines-12-00871-f003:**
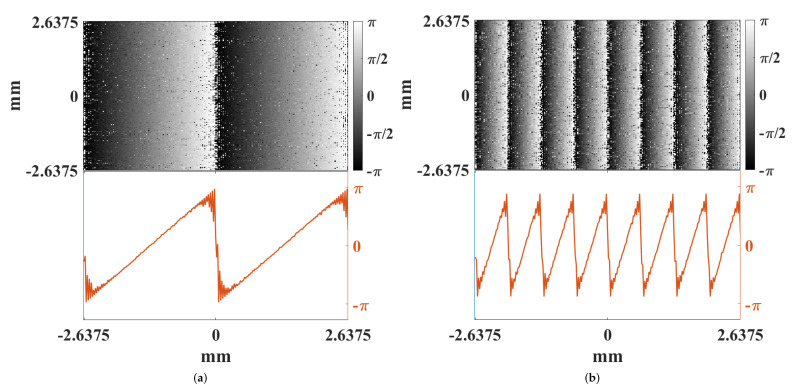
Computed phase tilts at the scatterer due to circular shifts of the SLM phase mask in the focal plane setup. The phase at the scatterer is simulated for a given phase mask, and then the phase mask is shifted, creating phase tilts at the scatterer. The SLM phase mask is circularly shifted 8 pixels in (**a**) and 32 pixels in (**b**). What is shown is the the phase differences at the scatterer’s surface between the unshifted and shifted cases.

**Figure 4 micromachines-12-00871-f004:**
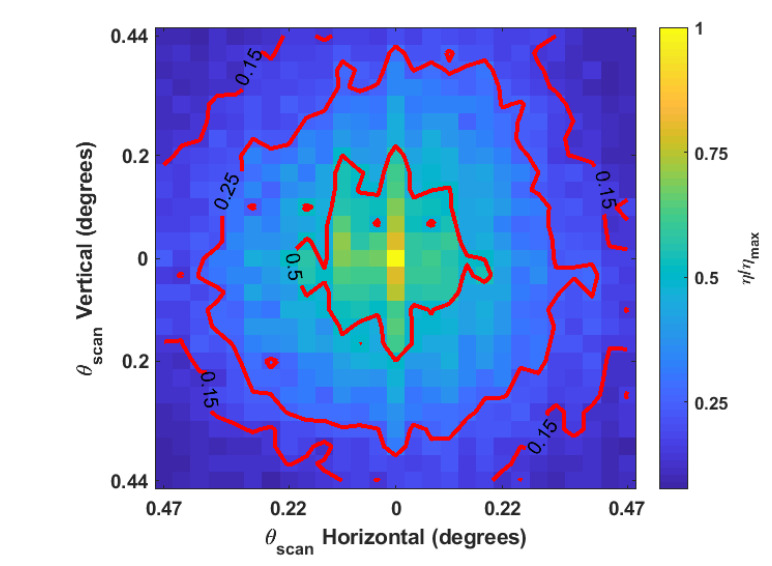
Experimental enhancement of displaced spot as a two-dimensional function of displacement size after a circular shift of the SLM phase map of a focal plane system shown in Figure 1 for 600-grit polished nickel. Results were averaged over five trials. The maximum value of η prior to normalization was 89.9±1.5.

**Figure 5 micromachines-12-00871-f005:**
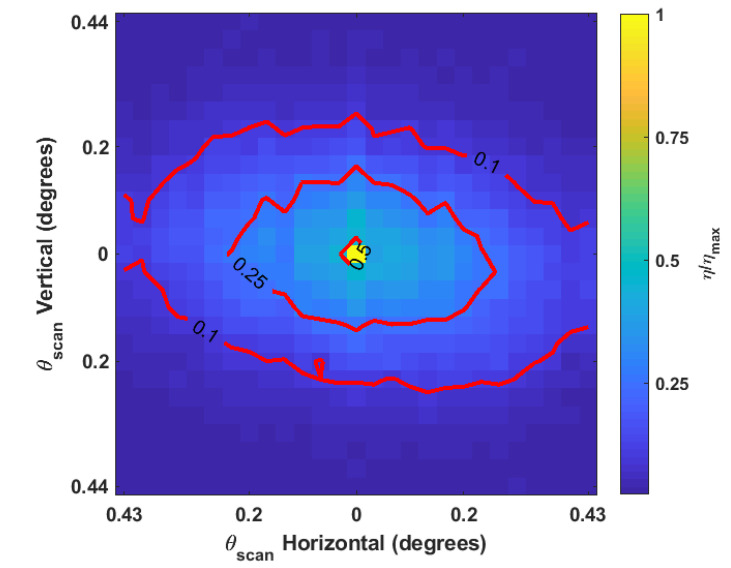
Experimental enhancement of displaced spot as a two-dimensional function of displacement size after a circular shift of the SLM phase map, of focal plane system shown in Figure 1 zinc oxide paint, on glass. Results were averaged over five trials. The maximum value of η prior to normalization was 80±3.

**Figure 6 micromachines-12-00871-f006:**
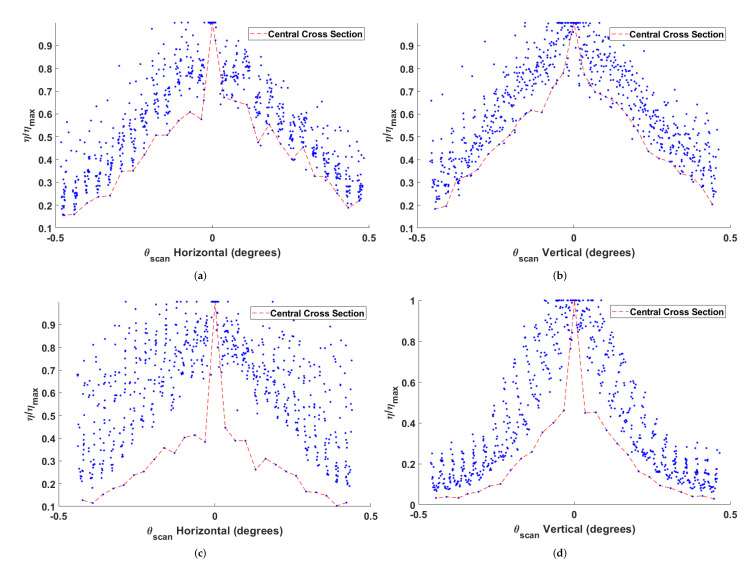
Scatterplot views of enhancement along the horizontal and vertical cross sections of Figure 4 and Figure 5, normalized to each cross section’s maximum central enhancement. The metal sample’s quasi-linear decrease for both horizontal and vertical scans is shown in (**a**,**b**). The paint sample’s decrease is shown in (**c**,**d**). In each scatterplot, the profile of the cross section passing through the overall center maximum is identified.

**Table 1 micromachines-12-00871-t001:** Surface properties of scattering samples, from a profilometer. Roughness is the standard deviation of surface height. $l_{c}$ is the distance required to shift the autocorrelation of the surface height profile by 1/e. Slope is the RMS surface slope. $l_{\lambda /2}$ is the distance required to create a height change of λ/2 assuming the surface has a linear slope with the value of the RMS slope.

Sample	Roughness	Correlation, $l_{c}$	Slope	Correlation $l_{\lambda /2}$
600-grit Nickel, average	0.45μm	16μm	0.066	4.79μm
White Paint on Glass, average	0.71μm	20μm	0.086	3.69μm

**Table 2 micromachines-12-00871-t002:** Predicted and observed scan angle increments due to a 16-pixel SLM shift. These values are the angular increments of $\theta_{scan}$ between each grid square in Figure 4 and Figure 5. Because the SLM was shifted 16 pixels at a time, the actual minimum scan angle increment is $\frac{1}{16}$ these reported values.

Scan: Material, Axis	Equation (Equation 4) $\theta_{\mathrm{scan}}$ Increment	Measured $\theta_{\mathrm{scan}}$ Increment
Nickel, horizontal	0.0343°±0.0009°	0.0363°±0.0004°
Nickel, vertical	0.0344°±0.0009°	0.0340°±0.0006°
Paint, horizontal	0.0343°±0.0009°	0.0328°±0.0004°
Paint, vertical	0.0344°±0.0009°	0.0363°±0.0004°

## Data Availability

Data underlying the results presented in this paper are not publicly available at this time but may be obtained from the authors upon reasonable request.

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
