# Peer review of "Beam Formation and Vernier Steering Off of a Rough Surface"

_micromachines, 2021, doi:10.3390/mi12080871_

Round 1

Reviewer 1 Report

The authors provided a method of beam steering of the scattered light using a reflective inverse diffusion method. The genetic algorithm assisted iterative feedback-based wavefront shaping algorithm has been proposed. It is well organized and would be useful for advanced imaging and sensing technology. I would like to recommend acceptance once several minor issues are handled. 1. More various experiment can be conducted to verify the proposed algorithm like diffraction angle switching for beam scanning applications or focused spot change in 2D image plane. 2. There are several similar approaches dealing with scattering control in disordered media or in metasurfaces. (Phys. Rev. Lett. 113, 113901 (2014); Nat. Photonics 7, 454-458 (2013); Light: Sci. & Appl. 7, 63 (2018)) Please discuss those references and compare them with your approach. 3. Such technology could be useful for future nanophotonic LiDAR system. (Nat. Nanotechnol. 16, 508-524 (2021)) Please discuss application feasibility of the proposed beam steering method in LiDAR.

Reviewer 2 Report

In this manuscript, Nagamine et al. describe a verification of wavefront shaping in reflection geometry that reproduces the results of previous works. I have a few questions about their experiments.

  1. The experimental geometry intended seems to be where the SLM is located at the Fourier plane with respect to the ‘scattering media = lens’. However, Z1 and f of L1 in their setup are not the same. This means that the Fourier relationship between the SLM and the field incident of the scattering media does not hold correctly. What’s the reason for this incorrect alignment?
  2. In lines 90 – 110, the parameters for wavefront shaping are described. Why are the detector channels sizes different for the CSA?
  3. In figure 2, CSA does not work at all showing that experimental conditions are not optimal. I agree that this can show that genetic algorithm is more robust as demonstrated in previous works, but the validity of the presented data is questionable.
  4. If a circular shift is used on the SLM plane, different parts of the incident beam will be controlled with respect to the shift. In other words, if the shift is large, you won’t have any wavefront shaped light going into the scattering sample. (The corrected part of the SLM is shifted out of the beam). If the objective is to scan or steer the reflected light, why not just place the SLM in the conjugate image plane? In this case you can just display a linear phase tilt directly on the SLM which has been shown in previous works.

Round 2

Reviewer 2 Report

2. The authors state 'It is clarified in lines 76-77 that the diffraction limited spot size is inversely proportional to the number of SLM segments. ' It would be helpful to the readers if experimental proof is given for this statement. The spot size will be decided by the area of diffuse reflection on the scattering surface.

3. Did the authors check the effect of the circular shift on the SLM and how the incorrect wavefront correction per shift impacts the results? The reply doesn't answer my question. 

In addition, the authors state that ' Finally, when steering with an imaging plane system, the light off of the SLM will itself be steered when 
phase tilts are applied. This creates the possibility of steering the beam too far off-axis on the lens.' This is almost impossible. The slm used in the paper has 15 um pixel size which can only tilt the wavefront to about 2 degrees max.
